# CCL18 Expression Is Higher in a Glioblastoma Multiforme Tumor than in the Peritumoral Area and Causes the Migration of Tumor Cells Sensitized by Hypoxia

**DOI:** 10.3390/ijms23158536

**Published:** 2022-08-01

**Authors:** Szymon Grochans, Jan Korbecki, Donata Simińska, Wojciech Żwierełło, Sylwia Rzeszotek, Agnieszka Kolasa, Klaudyna Kojder, Maciej Tarnowski, Dariusz Chlubek, Irena Baranowska-Bosiacka

**Affiliations:** 1Department of Biochemistry and Medical Chemistry, Pomeranian Medical University in Szczecin, Powstańców Wlkp. 72, 70-111 Szczecin, Poland; szymongrochans@gmail.com (S.G.); jan.korbecki@onet.eu (J.K.); d.siminska391@gmail.com (D.S.); dchlubek@pum.edu.pl (D.C.); 2Department of Ruminants Science, Faculty of Biotechnology and Animal Husbandry, West Pomeranian University of Technology, Klemensa Janickiego 29 St., 71-270 Szczecin, Poland; 3Department of Medical Chemistry, Pomeranian Medical University in Szczecin, Powstańców Wlkp. 72, 70-111 Szczecin, Poland; wojciech.zwierello@pum.edu.pl; 4Department of Embryology and Histology, Pomeranian Medical University in Szczecin, Powstańców Wlkp. 72, 70-111 Szczecin, Poland; sylwiazeszotek@pum.edu.pl (S.R.); agnieszka.kolasa@pum.edu.pl (A.K.); 5Department of Anaesthesiology and Intensive Care, Pomeranian Medical University in Szczecin, Unii 13 Lubelskiej 1, 71-281 Szczecin, Poland; klaudynakojder@gmail.com; 6Department of Physiology, Pomeranian Medical University in Szczecin, Powstańców Wlkp. 72, 70-111 Szczecin, Poland; maciej.tarnowski@pum.edu.pl

**Keywords:** CCL18, PITPN3, CCR8, hypoxia, glioblastoma multiforme, tumor-associated macrophage, brain tumor

## Abstract

Glioblastoma multiforme (GBM) is a brain tumor with a very poor prognosis. For this reason, researchers worldwide study the impact of the tumor microenvironment in GBM, such as the effect of chemokines. In the present study, we focus on the role of the chemokine CCL18 and its receptors in the GBM tumor. We measured the expression of CCL18, CCR8 and PITPNM3 in the GMB tumor from patients (16 men and 12 women) using quantitative real-time polymerase chain reaction. To investigate the effect of CCL18 on the proliferation and migration of GBM cells, experiments were performed using U-87 MG cells. The results showed that CCL18 expression was higher in the GBM tumor than in the peritumoral area. The women had a decreased expression of PITPNM3 receptor in the GBM tumor, while in the men a lower expression of CCR8 was observed. The hypoxia-mimetic agent, cobalt chloride (CoCl_2_), increased the expression of CCL18 and PITPNM3 and thereby sensitized U-87 MG cells to CCL18, which did not affect the proliferation of U-87 MG cells but increased the migration of the test cells. The results indicate that GBM cells migrate from hypoxic areas, which may be important in understanding the mechanisms of tumorigenesis.

## 1. Introduction

Glioblastoma multiforme (GBM) is the most severe of brain tumors at grade IV [1]. Compared to other cancers, while its incidence is low at approximately 3.2 per 100,000 population [2], and despite the use of surgery, radiotherapy or chemotherapy, GBM patients show a very short median survival prognosis following diagnosis and treatment of only about 12 months [3,4] and a 5-year survival rate of less than 5% [2,5]. For this reason, GBM is being intensively studied to better understand the tumorigenesis-related mechanisms of this cancer, to help develop more effective therapeutic approaches.

One of the current key directions in the study of tumorigenic processes in GBM tumors is to understand the communication between tumor cells, and between a tumor cell and tumor-associated cells, where chemokines play an important role. The chemokine group of approximately 50 chemotactic cytokines [6] plays significant roles in the migration, invasion and metastasis of tumor cells, as well as in the angiogenesis and recruitment of tumor-associated cells into the tumor niche [7].

The expression of some chemokines is elevated in cancer tumors, with GBM showing elevated expressions of 17 chemokines in the tumor compared to healthy brain tissue [8]. The greatest increase in expression (at 70 times more, relative to controls) was noted for CC motif chemokine ligand 18 (CCL18) [8], from the β-chemokine sub-family. In cancer tumors, CCL18 is mainly produced and secreted into the tumor microenvironment by tumor-associated macrophages (TAM) [9,10,11,12,13,14] and in much smaller amounts by GBM cancer cells [11], as shown in non-small cell lung cancer cells [12] and melanoma cells [15].

The best-researched CCL18 receptor is phosphatidylinositol transfer membrane-associated protein 3 (PITPNM3)/PYK2 N-terminal domain interacting receptor 1 (Nir1) [16,17,18,19,20]. Activation of this receptor causes tumor cell migration in many cancers. Activation of PITPN3 by CCL18 on endothelial cells also leads to angiogenesis [21]. No less a significant receptor for CCL18 in cancer processes is CC motif chemokine receptor 8 (CCR8). In bladder cancer, activation of CCR8 by CCL18 causes the migration and invasion of tumor cells, as well as increased expression of vascular endothelial growth factor (VEGF)-C, which then causes lymphangiogenesis [22]. The CCL18→CCR8 axis has previously been shown to be significant in other cancer processes as well. CCL18 is involved in the availability of extracellular vesicles [23]. However, the significance of CCR8 in the pro-tumorigenic effects of CCL18 has not yet been sufficiently investigated.

Another important direction of cancer research is to understand the impact of hypoxia on cancer processes [24,25]. Hypoxia is an oxygen deficiency characteristic of certain areas of a cancerous tumor. The cellular response to hypoxia is associated with activation of the hypoxia-inducible factor (HIF)-1 and HIF-2 pathways [26]. Oxygen deficiency decreases the activity of the oxygen-dependent enzymes prolyl hydroxylase (PHD) [27,28] and factor-inhibiting HIF (FIH) [29,30]. As a consequence, HIF-1 and HIF-2 subunits are not proteolytically degraded. This accumulation in the cell is followed by the production of HIF-1 and HIF-2 and an increase in hypoxia-dependent gene expression. One of the model genes induced by HIF-1 (and thus by hypoxia) is vascular endothelial growth factor (VEGF) [31,32,33,34]. Hypoxia also alters gene expression independently of HIFs. This is associated with the reduced activity of the oxygen-dependent enzymes ten-eleven translocation (TET) and Jumonji C family (JmjC), which respectively lead to DNA and histone methylation [35].

The effect of CCL18 has been fairly well established in various cancer models. In comparison, very little research has been devoted to the expression of CCL18 receptors, in particular PITPNM3. Therefore, the aim of the present study was to investigate the contributions of CCL18 and the receptors CCR8 and PITPNM3 to tumorigenic processes in GBM. In particular, we focused on the importance of hypoxia and patient gender in the studied mechanisms of tumorigenic processes.

## 2. Results

### 2.1. CCL18 Expression in GBM Tumors Is Elevated Relative to the Peritumoral Area

CCL18 expression in the enhancing tumor region and the tumor core was more than seven times higher than in the peritumoral area (Figure 1). These differences were statistically significant (enhancing tumor region vs. peritumoral area, *p* = 0.044; tumor core vs. peritumoral area, *p* = 0.018). The expression of the PITPNM3 and CCR8 receptors did not differ in enhancing tumor region and the tumor core vs. peritumoral area (*p* > 0.05).

CCL18 expression in GBM tumors was statistically significantly higher in women (*p* < 0.05) in both the enhancing tumor region and the tumor core relative to the peritumoral area, but otherwise did not differ between the genders (Figure 2).

Expression of PITPNM3 in women was lower in the tumor core relative to both the enhancing tumor region (*p* = 0.049) and peritumoral area (*p* = 0.049). Expression of PITPNM3 in the tumor core in the women was also lower than in the men (*p* = 0.0011), but not statistically significantly (*p* = 0.062) in the enhancing tumor region in the women.

CCR8 expression in the peritumoral area and enhancing tumor region did not differ between genders. CCR8 expression in the tumor core in men was lower than in the peritumoral area (*p* = 0.0033) and lower in the tumor core compared to the women (*p* = 0.0077).

### 2.2. Hypoxia Increases the Expression of CCL18 and the Receptor for This Chemokine: PITPNM3 in U-87 MG Cells

The hypoxia-mimetic agent CoCl_2_ increased the expression of CCL18 (*p* < 0.0001), PITPNM3 (*p* < 0.0001) and VEGF expression (*p* = 0.002) in U-87 MG cells (Figure 3). There was no effect of cobalt chloride (CoCl_2_) on CCR8 expression (*p* > 0.05). Nutrient deficiency did not affect the expression of CCL18 or its receptors, PITPNM3 and CCR8 (*p* > 0.05), but did decrease VEGF expression (*p* = 0.0007) in U-87 MG cells.

### 2.3. CCL18 Does Not Affect U-87 MG Cell Proliferation

CCL18 had no effect on U-87 MG cell proliferation at concentrations ranging from 10 ng/mL to 50 ng/mL (*p* > 0.05) (Figure 4).

### 2.4. CCL18 Induces Migration of U-87 MG Cells. This Effect Is Enhanced by Preincubation with CoCl_2_

CCL18 increased U-87 MG cell migration at concentrations ranging from 10 ng/mL to 50 ng/mL (*p* < 0.01) (Figure 5 and Figure 6). At a concentration of 10 ng/mL, CCL18 increased cell migration by 36%, while at a concentration of 50 ng/mL it increased expression three-fold. This effect was synergistic with preincubation of U-87 MG cells with CoCl_2_ where it increased the number of migrating cells four-fold. U-87 MG line cells showed migration to CCL18, which was further stimulated by the hypoxia-mimetic agent CoCl_2_.

### 2.5. Macrophages and CCL18 Expression Are Co-Localized with Each Other in the GBM Tumor

In tumor-transformed brain tissue, there were areas with many macrophages (Figure 7, red arrows) that were metabolically active to synthetize and liberate CCL18 into ECM of glioblastoma (Figure 7, green arrows) within the region where hypoxia and malnutrition occurred.

## 3. Discussion

### 3.1. In Vivo Expression of the Studied Genes

In our study, we demonstrated that the expression of the chemokine CCL18 is elevated in GBM tumors. These results are consistent with literature data, where the expression of CCL18 in GBM tumor can be 70 times higher [8] to 100 times higher [36] than in post-mortem brain biopsies. CCL18 plays an important role in tumorigenesis in GBM, as shown by the survival data of patients with this cancer which is inversely correlated with CCL18 levels in the tumor [23]. Additionally, data from the Human Protein Atlas (Available online: http://www.proteinatlas.org, accessed on 22 August 2021) [37,38,39] show that patients with high CCL18 expression in glioma tumors have a much worse prognosis.

Our study is the first to analyze the expression of the receptor for the CCL18 chemokine, i.e., PITPNM3 in GBM. Therefore, it cannot be directly compared to the literature data. We showed that the expression of PITPNM3 did not change in GBM tumors in all studied patients. However, when taking gender into account, it did decrease in women compared to the peritumoral area. The influence of the PITPNM3 receptor on tumorigenic processes has been best studied in breast cancer models, which showed an elevated expression of this receptor [16,40], as well as non-small cell lung cancer [20] and oral squamous cell carcinoma [41]. It has been reported that this receptor may play an important role in the development of some cancers. Data included from the Human Protein Atlas (Available online: http://www.proteinatlas.org, accessed on 22 August 2021) [37,38] indicate that higher expression of PITPNM3 in a tumor is associated with a worse prognosis for breast cancer patients [42]. The same regularity can be observed for glioma [43]. Hence, PITPNM3 appears to play a key role in GBM development. Our results may explain the higher incidence and mortality of male patients with GBM [44,45], although there is a lack of literature data on the influence of gender on PITPNM3 expression.

In our study, CCR8 expression did not change relative to peritumoral area. However, there was some trend of decreased CCR8 expression in the necrotic core relative to peritumoral area. CCR8 expression decreased in the necrotic core in male patients. Previous literature has shown that CCR8 expression in GBM tumors is either unchanged [8] or increased [23]. CCR8 is important in the development of GBM because, as previously shown, patients with increased CCR8 expression in the tumor show a worse prognosis [23]. These results are in line with data presented in the Human Protein Atlas (Available online: http://www.proteinatlas.org, accessed on 22 August 2021) [37,38,46].

### 3.2. Effects of Hypoxia, Oxidative Stress and Nutrient Deficiency Condition on CCL18 Expression and Function

In our model, hypoxia increased the expression of VEGF. This shows that the experiment was properly performed because VEGF is a model gene that is inducible by hypoxia and by CoCl_2_ [32,34,47]. In our study, nutrient deficiency induced the downregulation of VEGF expression, which is consistent with the results of Sarkar et al. (2020) who showed a reduction in VEGF expression in MG-63 osteosarcoma cells in response to reduced fetal bovine serum (FBS) concentration [48]. GBM tumors are also characterized by a decrease in nutrient availability in hypoxic zones. Both nutrient deficiency and hypoxia affect VEGF expression in GBM cells in hypoxia-affected zones.

In our study, hypoxic conditions increased CCL18 expression in U-87 MG cells. However, the observed effect may depend on the research model used. In non-neoplastic cells, hypoxia reduces CCL18 expression, as shown in immature dendritic cells [49,50], monocytes [51] and macrophages [52]. This is associated with a decrease in JMJD3 activity and consequently an increase in histone methylation in the *CCL18* gene promoter. As the CCL18 gene promoter does not contain a hypoxia-response element (HRE), it is therefore not induced by HIF-1 or HIF-2 [50]. In contrast, in lung adenocarcinoma cells, hypoxia reduces CCL18 expression, but the observed effect was not statistically significant [53].

Our results show that the expression of PITPNM3 is higher under hypoxia, which is consistent with the literature data. PITPNM3 is a gene that undergoes increased expression under hypoxia. This is confirmed by studies on PC-3 prostate cancer cells and SK-OV-3 ovarian cancer cells [33].

In our study, hypoxia did not alter CCR8 expression in U-87 MG cells. Currently, there is no data available on the effect of hypoxia on the expression of the CCR8 receptor, which supports the results obtained in whole transcriptome analysis using microarrays using three cell lines, in which no effect of hypoxia on CCR8 expression was demonstrated [33].

### 3.3. Effect of CCL18 on U-87 MG Cell Proliferation

In the present study, CCL18 had no effect on U-87 MG cell proliferation. These results are partially consistent with the literature data. As shown, CCL18 did not affect the proliferation of BxPC-3 and PANC-1 pancreatic ductal adenocarcinoma cells [54], MGC-803 gastric cancer cells and GES-1 gastric epithelial cells [55]. Nevertheless, the effect of CCL18 seems to depend on the research model. For example, CCL18 has been shown to increase the proliferation of U-251 GBM cells [36], MDA-MB-231 and MCF-7 breast cancer cells [56]. It is likely that the effect of CCL18 on proliferation depends on the expression of receptors and other proteins significant in the induction of proliferation by this chemokine.

### 3.4. Effect of CCL18 on U-87 MG Cell Migration

In our study, CCL18 induced the migration of U-87 MG cells. These results support literature data showing that CCL18 causes the migration and invasion of breast cancer cells [16], bladder cancer cells [22], hepatocellular carcinoma cells [19], non-small cell lung cancer cells [20] and prostate cancer cells [17]. Our study demonstrated for the first time that hypoxic conditions increase cancer cell migration. Our results have significant implications for understanding cancer tumor function, particularly in GBM, where hypoxia-affected zones which form during tumor development induce cancer cell migration.

## 4. Materials and Methods

### 4.1. Patient Samples

The material used in the present study was obtained from patients under surgery for brain tumors who had been diagnosed by neuroimaging (magnetic resonance imaging (MRI) or computed tomography (CT)) at the Department of Neurosurgery and Pediatric Neurosurgery of the Pomeranian Medical University in Szczecin, Poland. The present project and archiving of material were initiated in 2014 by the Department of Biochemistry and the Department of Neurosurgery and Pediatric Neurosurgery of the Pomeranian Medical University in Szczecin and concerned the engagement of purinergic receptors in GBM progression. The project was accepted by the local bioethical commission (KB-0012/96/14) and the study was conducted in accordance with the Declaration of Helsinki.

Tissue samples from tumors were collected during surgery from 28 patients (16 males and 12 females) diagnosed with a central nervous system (CNS) tumor and GBM (Table 1 and Figure 8). All patient brain tumor samples were analyzed in 6 replicates. Patients presented with symptoms resulting from increased intracranial pressure: dizziness andnausea, and those resulting from local tumor growth: sensory and motor disorders as well as disorders of higher nervous functions (Table 2).

Each patient was recommended for neurosurgery following radiological diagnosis of a CNS tumor. After qualifying for surgery, patients underwent a standard anesthetic procedure (general anesthesia with endotracheal intubation). During the neuronavigation procedure, craniotomy and tumor resection were performed according to the classical method (bone removal and dura incision, tumor visualization, resection, biopsy for histopathological and molecular examination, closure of the dura, bone restoration in some patients, subcutaneous tissue and skin closure in some patients). The range of resection was determined by the extent of the tumor and its topography.

Clinical radiological morphology made it possible to distinguish three tumor zones commonly found in the literature and in clinical practice: the non-enhancing tumor core (TC) (usually located in the central part of the tumor), the enhancing tumor region (ET) (surrounding the tumor core) and the peritumoral area (PA) (a buffer zone between the tumor and healthy tissue, with individual foci of infiltration) (Figure 9) [57,58]. We considered the peritumoral area as the experimental control; as previously shown, this is a suitable control for GBM-related experiments [59].

The use of neuronavigation helped in mapping the tumor and determining the topographies of the zones. The results of MRI scans at both 1.5 T and 3 T were entered into the control station of the neuronavigation device and used during the operation to determine the position of surgical instruments in relation to cancer tissue with the assistance of a video camera. The camera monitored the surgical movement in relation to the radiological image to a precision of 2–3 mm. This allowed a safe and reliable resection in places where the image of the operating microscope was uncertain and the macroscopic tumor boundaries were difficult to distinguish. Neuronavigation during biopsy and craniotomy allowed material to be extracted from the three separate zones. Each sample was subjected to histopathological examination to confirm the criteria of a grade IV brain tumor defined by WHO: *IDH* mutation, 1p19q codeletion and *MGMT* gene promoter methylation.

### 4.2. Cell Culture and Treatment

Human brain cells (glioblastoma astrocytoma, U-87 MG cell line) from the European Collection of Authenticated Cell Cultures (ECACC) were cultured in EMEM medium (Sigma-Aldrich, Poznań, Poland) supplemented with 10% (*v*/*v*) heat-inactivated fetal bovine serum (FBS; Gibco Limited, Poznań, Poland), 2 mM L-glutamine, 1 mM sodium pyruvate (Sigma-Aldrich, Poznań, Poland), 1% non-essential amino acids (Sigma-Aldrich, Poznań, Poland), 100 U/mL penicillin (Gibco Limited, Poznań, Poland) and 100 µg/mL streptomycin (Gibco Limited, Poznań, Poland), at 37 °C in a humidified atmosphere of 95% air and 5% CO_2_. The U-87 MG cells were seeded in 6-well plates at a density of 20,000 cells/cm^2^ in full medium. After 72 h of incubation (70–80% confluence), the cells were washed three times with pre-warmed phosphate buffer saline (PBS) solution (37 °C). Next, the cells were cultured for 24 h under three different conditions (control, nutrient deficient and necrotic). The control cells were suspended in a full medium, the starved cells were grown in a medium with a low concentration of L-glutamine (0.2 mM) and without sodium pyruvate (volume supplemented with PBS). For the induction of necrotic conditions, cells were incubated in a medium supplemented with 200 µM H_2_O_2_. After 24 h of incubation the U-87 MG cells were trypsinized (0.25% trypsin-EDTA solution, Sigma-Aldrich, Poznań, Poland) from the plate. After centrifuging (25 °C, 300 G, 5 min), the supernatant was discarded, the pellet rinsed with PBS and centrifuged again for RNA analysis.

In vitro studies were performed to analyze the influence of hypoxia and nutrient deficiency condition on the gene expression of *CCL18, CCR8* and *PITPNM3*. All cell culture studies were analyzed in 6 replicates in each of the study groups.

Characteristic of GBM is the presence of structures called pseudopalisades in the tumor [60,61,62]. These structures exhibit hypoxia and nutrient deficiency associated with the long distance or blocked lumen of blood vessels running through the pseudopalisade. To analyze the effect of conditions in pseudopalisades on GBM cells, we examined the effect of hypoxia and nutrient deficiency on the expression of *CCL18, CCR8* and *PITPNM3* genes. For this purpose, the U-87 MG cells were treated with cobalt chloride (CoCl_2_) (200 mM, Poznan, Poland), a hypoxia-mimetic agent widely used in hypoxia experiments [63,64]. In order to demonstrate properly performed hypoxia-mimetic conditions, the effect of CoCl_2_ on *VEGF* expression was investigated. *VEGF* is a model gene whose expression is upregulated under hypoxia, such as in incubation with CoCl_2_ [32,34,47].

In addition, to better reflect the conditions in which GBM cells live, we studied the effect of nutritional deficiency on the expression of *CCL18*, *CCR8* and *PITPNM3* in U-87 MG cells. These cells grew in a medium with a low concentration of L-glutamine (0.2 mM) and without sodium pyruvate. However, they were still exposed to 1.0 g/L (5.5 mM) of glucose in the medium. Under these conditions, the concentration of substances such as mineral salts, vitamins, other amino acids or growth factors did not change.

The aforementioned experimental conditions reflect nutritional deficiency conditions. GBM cells have two sources of energy and building blocks: glucose and glutamine [65,66]. GBM cells convert glucose to pyruvate, which is then metabolized in the Crebs cycle or is converted to lactate. The reduction in pyruvate decreases the availability of glucose already metabolized. GBM cells also need a second component for their metabolism—glutamine [67,68], a source of carbon for proliferating GBM cells. A significant decrease in the concentration of glutamine is part of a nutritional deficiency condition.

### 4.3. Quantitative Real-Time Polymerase Chain Reaction (qRT-PCR)

Quantitative analysis of the mRNA expression of *CCL18, CCR8* and *PITPNM3* genes was performed by two-step reverse transcription PCR (RT-PCR). Total RNA was extracted from 50–100 mg tissue samples using an RNeasy Lipid Tissue Mini Kit (Qiagen, Hilden, Germany) and, for the in vitro study, from 300,000 cells using an RNeasy Mini Kit. cDNA was prepared from 1 μg of total cellular RNA in 20 μL of reaction volume using a FirstStrand cDNA synthesis kit and oligo-dT primers (Fermentas, Waltham, MA, USA). Quantitative assessment of mRNA levels was performed using an ABI 7500Fast real-time RT-PCR analyzer with Power SYBR Green PCR Master Mix reagent (Applied Biosystems, Waltham, MA, USA). Real-time conditions were 95 °C (15 s), 40 cycles at 95 °C (15 s), and 60 °C (1 min). According to melting point analysis, only one PCR product was amplified under these conditions. Each sample was analyzed in two technical replicates, and mean Ct values were used for further analysis. The relative quantity of a target, normalized to the levels of endogenous controls, *glyceraldehyde-3-phosphate dehydrogenase*
*(GAPDH)*, was calculated as the fold difference (2^dCt^) and further processed using statistical analysis. Data were presented as tumor tissue absolute expression.

The *GAPDH* reference gene was selected because it is considered a suitable control in research on the expression of various genes in GBM [69]. The following primer pairs were used:

(5′-TCA TGG GTG TGA ACC ATG AGA A-3′ and 5′-GGC ATG GAC TGT GGT CAT GAG-3′) for *GAPDH*,

(5′-CTCTGCTGCCTCGTCTATACCT-3′ and 5′-CTTGGTTAGGAGGATGACACCT-3′) for *CCL18,*

(5′-GTGTGACAACAGTGACCGACT-3′ and 5′-CTTCTTGCAGACCACAAGGAC-3′) for *CCR8*,

(5′-TCGCTTGTCTCACCTGAAC-3′ and 5′-CAGGAACTCTCTGTAGACCTGG-3′) for *PITPNM3*.

### 4.4. Proliferation

U-87 MG cells were seeded in a 96-well plate at a density of 20,000 cells/cm^2^ in a complete medium. After 24 h of incubation, the medium was drawn off and the cells were washed with warm PBS. Then, full medium with either 10 ng/mL, 20 ng/mL or 50 ng/mL of CCL18 (SigmaAldrich, Poznań, Poland) was added to the wells. For each given concentration of CCL18 under study, 8 replicates were performed. The control for each concentration was a sample (8 replicates) with an appropriate amount of PBS added so that the final concentrations of the medium components in the compared samples were equivalent to each other. Cells were incubated in the provided medium for 48 h. Then 20 µL of 3-(4,5-dimethylthiazol-2-yl)-2,5-diphenyltetrazolium (MTT) (5 µg/µL, SigmaAldrich, Poznań, Poland) was added to each well and incubated for 2 h in an incubator. In the next step, the culture medium was gently and thoroughly removed and 150 μL of dimethyl sulfoxide (DMSO) (SigmaAldrich, Poznań, Poland) was added to each well and incubated in the dark for 10 min. The absorbance of each well was measured using a microplate reader (EZ Read 2000, Biochrom, Poland) at 590 nm.

### 4.5. Measuring Cell Confluence Using ImageJ

Micrographs (Leica DM5000B, Wetzlar, Germany, magnification ×4) of chambers from migration assay stained with hematoxyline were analyzed with the use of ImageJ Fiji software (Johannes Schindelin, Albert Cardona, Mark Longair, Benjamin Schmid, and others, https://imagej.net/software/fiji/downloads, version 1.2, accessed on 20 September 2021). First, the downloaded image was opened in Fiji software and color deconvolution was performed (Image > Color > Color Deconvolution), and the hematoxyline option was selected. Next, the threshold was selected (Image > Adjust > Treshold); the minimum threshold value was set at zero and the maximum threshold value was adjusted so that the background signal was removed, without removing the true signal from the hematoxyline stained cells. The percentage of area covered by the cells was measured (Analyze > Measure). Three independent analyses (with a slightly changed maximum threshold) were performed and the results were averaged.

### 4.6. Migration

U-87 MG cells were seeded in two 6-well plates at a density of 20,000 cells/cm^2^ and cultured for 48 h under different conditions. One was cultured in a complete medium with CoCl_2_ (200 mM, SigmaAldrich, Poznań, Poland) and the other without CoCl_2_. The medium was then extracted and the cells washed twice with warm phosphate buffer saline (PBS) solution. Full medium with CCL18 (SigmaAldrich, Poznań, Poland) at concentrations of 10 ng/mL, 20 ng/mL and 50 ng/mL, and medium with PBS was added to 5 wells of both plates so that the final concentration of medium components corresponded to the lowest in the other wells. Cells with a complete medium supplemented with CCL18 or PBS were incubated for 16 h. In the next step, the medium was extracted and the cells were washed twice with warm PBS, 1 mL of trypsin-EDTA solution (0.25%) was added to each well and incubated for 4 min until the cells detached. Complete medium was then added and centrifuged (25 °C, 300 G, 5 min). The resulting cell pellets were placed in a medium without FBS but with 1% bovine serum albumin (BSA) (SigmaAldrich, Poznań, Poland). Cells were counted for each well, and the corresponding volume of cell mixtures along with the corresponding volume of medium without FBS but with 1% BSA was added to the upper chamber of Nunc™ Polycarbonate Cell Culture Inserts in multi-well plates (8.0 UM PC, Life Technologies, Warsaw, Poland) obtaining 1 × 10^5^ cells suspended in 300 μL of medium without FBS but with 1% bovine serum albumin (BSA). 750 μL of medium with 20% FBS was added to the lower chamber. The cells were incubated for 8 h. In a further step, a procedure was performed to fix the cells that migrated through the membrane. For this purpose, Polycarbonate Cell Culture Inserts were washed twice with PBS, then cells were fixed with formaldehyde solution (4%, buffered, pH 6.9) (SigmaAldrich, Poznań, Poland) for 3 min at 25 °C. The Polycarbonate Cell Culture Inserts were washed twice with PBS. In all, 100% methanol (Honeywell, Warsaw, Poland) was added to both chambers and incubated for 20 min at 25 °C. The Polycarbonate Cell Culture Inserts were again washed twice with PBS. The cells were then stained with hematoxylin (SigmaAldrich, Poznan, Poland) for 2 min at 25 °C, the chamber was rinsed twice with PBS, and then the top layer of cells from the Polycarbonate Cell Culture Inserts was wiped off leaving only the cells that had migrated the membrane. To obtain the results of the experiment, the Polycarbonate Cell Culture Inserts were placed on primary slides and the number of cells in each sample was counted using a microscope (Leica DMi1, KAWA.SKA, Poznań, Poland).

### 4.7. Immunohistochemistry

The dissected glioblastoma were fixed in 10% formalin for at least 24 h and then washed with absolute ethanol (3 times over 3 h), absolute ethanol with xylene (1:1) (twice over 1 h) and xylene (3 times over 20 min). Then, after 3 h of saturation of the tissues in liquid paraffin, the samples were embedded in paraffin blocks. Using a microtome (Microm HM340E), 3–5 μm serial sections were obtained and placed on polysine microscope slides (Thermo Scientific, Altrincham, UK; cat. no. J2800AMNZ). The sections of the glioblastoma were deparaffinized in xylene and rehydrated in decreasing concentrations of ethanol, and then used for IHC reaction.

In order to expose the epitopes to IHC procedure, the deparaffinized and rehydrated sections were boiled twice in Target Retrieval Solution (DacoCytomation, Carpinteria, CA, USA, S2367, S2369) in a microwave oven (700 W twice for 5 min). Once cooled and washed with PBS, the endogenous peroxidase was blocked using a 3% solution of perhydrol in methanol, and then the slides were incubated over night at 4 °C with primary antibodies against: CD68 (Abcam, Cambridge, UK, EPR20545, final dilution 1:5000), CCL18 (also now as MIP-4 from Santa Cruz Biotechnology, Dallas, TX, USA, sc-374438; final dilution 1:250). Antibodies were diluted in antibody diluent with background-reducing components (Dako, Santa Clara, CA, USA, S3022). To visualize the antigen–antibody complex, a Dako LSAB + System-HRP was used (DakoCytomation, K0679), based on the reaction of avidin–biotin–horseradish peroxidase with DAB as a chromogen, according to the staining procedure instructions included. Sections were washed in distilled H_2_O and counterstained with hematoxylin. For a negative control, specimens were processed in the absence of a primary antibodies. Positive staining was determined microscopically (Leica DM5000B, Wetzlar, Germany) by visual identification of brown pigmentation.

### 4.8. Statistical Methods

The expressions of the desaturase genes were calculated in relation to the expression of *GAPDH*. The relative expression values for the three zones (tumor core, enhancing tumor region and peritumoral area) in each patient and their ratios were calculated, e.g., enhancing tumor region/tumor core as a ratio of relative expression in the enhancing tumor region to the relative expression in the tumor core. The distribution of expression values significantly differed from a normal distribution (Shapiro–Wilk test), and therefore statistical analysis was based on nonparametric tests: Mann–Whitney U tests for comparisons between groups of patients, Wilcoxon signed-rank tests for comparisons between the zones of the tumor and in vitro studies. Spearman rank correlation coefficient was used for analysis of correlations between the expressions of the tested genes in the three zones. The median and quartile values were given as descriptive statistics in tables and graphs. The Kruskal–Wallis ANOVA test was used to test the demographic and basic characteristic of the study group regarding sex. The statistical significance threshold was *p* < 0.05. Calculations were performed using Statistica 13 software.

## 5. Conclusions and Significance of the Results Obtained

Macrophages accumulate in regions of the GBM affected by hypoxia [70]. The specific conditions in these regions polarize these cells toward M2 macrophages [71]. Macrophages with such polarization show increased expression of CCL18 [72,73]. CCL18 is even considered a kind of M2 marker of macrophages, and hence its first name: alternative macrophage activation-associated CC chemokine 1 (AMAC-1) [74]. Hypoxia causes increased expression of PITPNM3, a receptor for CCL18 in cancer cells [33], which was confirmed in the present study. The increase in CCL18 expression in the tumor cell environment and the increase in PITPNM3 receptor expression in the tumor cell simultaneously stimulate the migration of tumor cells. This mechanism contributes to GBM tumor dissemination and worsens the prognosis for patients with GBM. Experimental data also indicate that the survival of patients with a high expression of PITPNM3 and/or CCL18 have a worse prognosis than those with a lower expression of these genes [39,43], as shown by data from the Human Protein Atlas (Available online: http://www.proteinatlas.org, accessed on 22 August 2021) [37,38].

## 6. Limitations of the Study

This study showed that macrophages secrete CCL18, which increases the migration of GBM cells sensitized by hypoxia. It also demonstrated gender differences in the expression of receptors for CCL18. Nevertheless, it was associated with certain limitations. First, this study considered patients from a population from a relatively small region that is relatively genetically homogeneous, so the obtained results may differ from a similar study conducted elsewhere. For this reason, the experiment should be replicated in other countries to see whether its results were specifically related to the studied population or GBM in general. Second, the mechanism by which hypoxia sensitizes GBM cells to CCL18 should be investigated in greater detail. Our results indicate that hypoxia increases the expression of PITPNM3, the receptor for CCL18, which points to a potential mechanism in which hypoxia sensitizes GBM cells to CCL18 by increasing PITPNM3 expression. However, we did not focus on the importance of the PITPNM3 receptor in this process, but only on the potential role of CCL18 in GBM tumorigenesis. For this reason, future research should investigate the importance of PITPNM3 in GBM cell migration under hypoxia. Particularly useful could be experiments using siRNA or shRNA targeting PITPNM3 and CCR8 receptors.

## Figures and Tables

**Figure 1 ijms-23-08536-f001:**
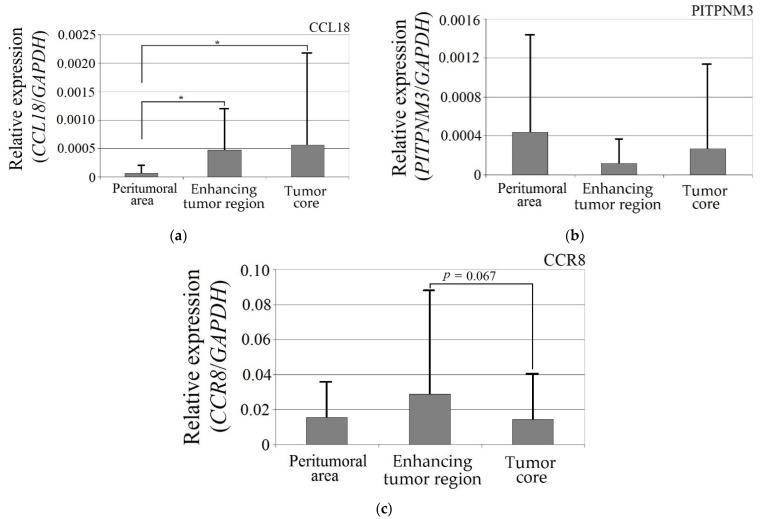
Expression of (**a**) CCL18, (**b**) PITPNM3 and (**c**) CCR8 in GBM tumor and peritumoral area. * *p* = 0.05.

**Figure 2 ijms-23-08536-f002:**
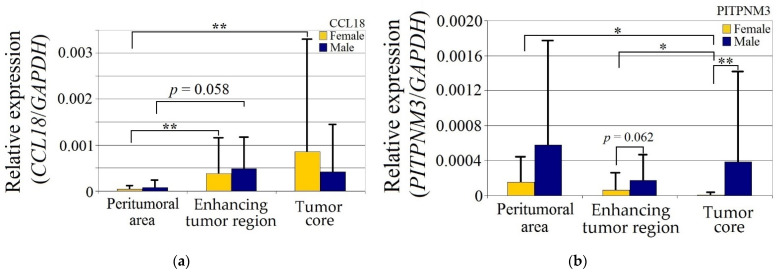
Expression of (**a**) CCL18, (**b**) PITPNM3 and (**c**) CCR8 in GBM tumor and peritumoral area relative to patient gender. * *p* = 0.05, ** *p* = 0.005.

**Figure 3 ijms-23-08536-f003:**
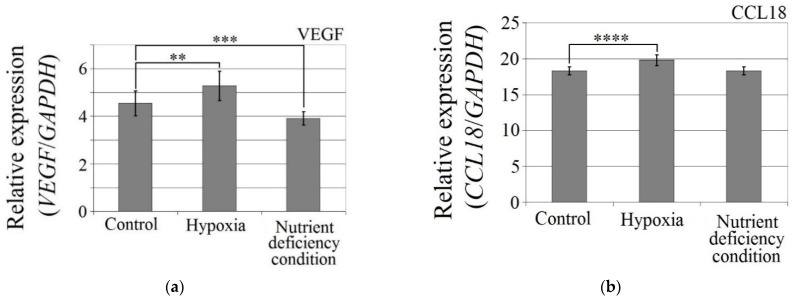
Effects of the hypoxia-mimetic agent CoCl_2_ and nutrient deficiency on the expressions of (**a**) VEGF, (**b**) CCL18, (**c**) PITPNM3 and (**d**) CCR8 in U-87 MG cells. ** *p* = 0.005, *** *p* = 0.001, **** *p* = 0.0001.

**Figure 4 ijms-23-08536-f004:**
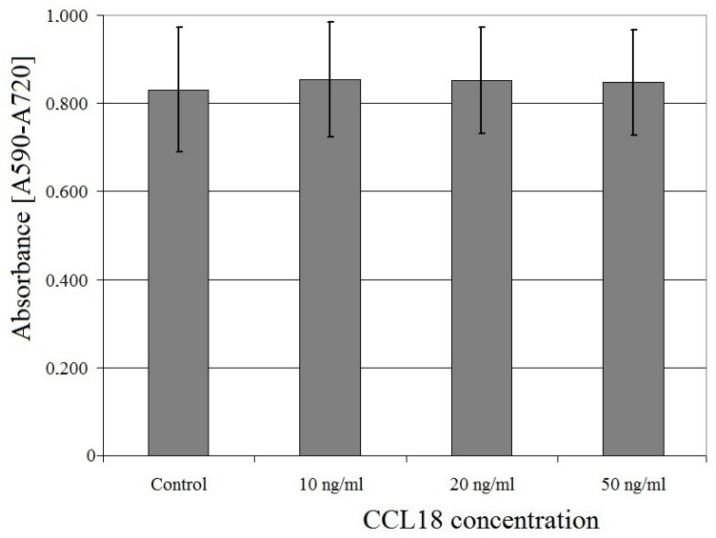
Effect of CCL18 on U-87 MG cell proliferation.

**Figure 5 ijms-23-08536-f005:**
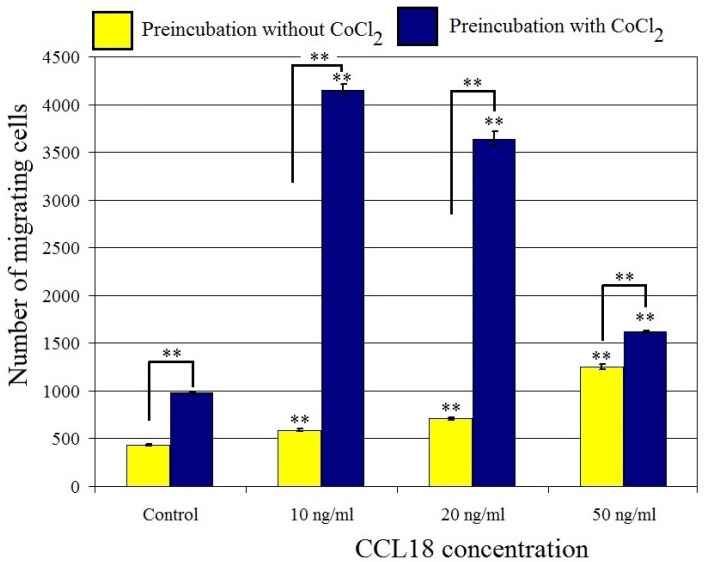
Effect of CCL18 on the migration of U-87 MG cells. ** *p* = 0.005.

**Figure 6 ijms-23-08536-f006:**
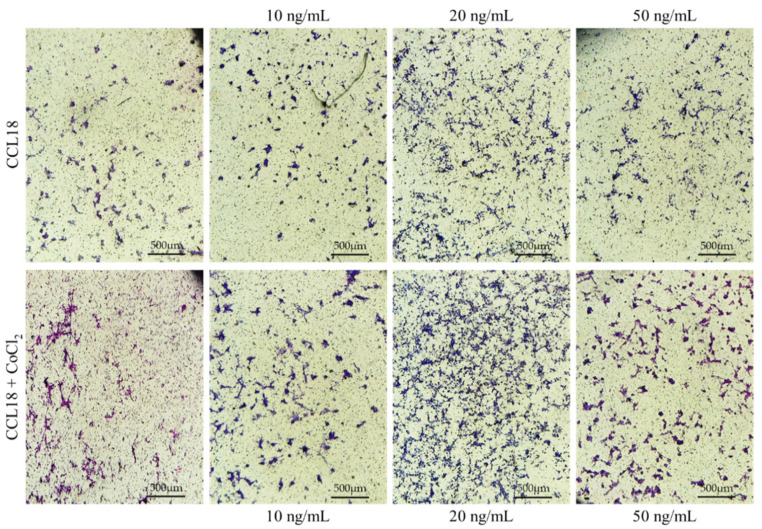
Representative micrographs showing the migration assay. U-87 MG cells cultured in CoCl_2_ hypoxia mimic agent (lower panel) showed a stronger response to CCL18 compared to cells grown under normal conditions (upper panel). CCL18 concentrations were 10 ng/mL, 20 ng/mL and 50 ng/mL. Control medium contained PBS. Pictures were captured with a Leica DM5000B, Wetzlar, Germany, objective magnification ×4, scale bar 500 µm.

**Figure 7 ijms-23-08536-f007:**
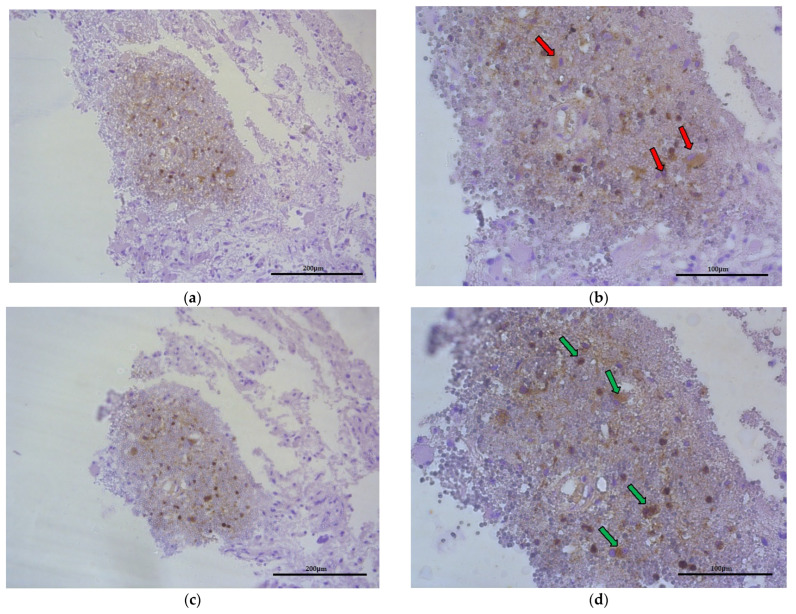
Representative microphotography showing the same region of tumor (obj. magnification a, b × 20, scale bar 200 µm; a, b × 40, scale bar 100 µm). (**a**,**b**) Immunoexpression of CD68 (marker of macrophages), red arrows indicate this cells. (**c**,**d**) Immunoexpression of CCL18, green arrows show that macrophages produce and liberate CCL18 into ECM of glioblastoma.

**Figure 8 ijms-23-08536-f008:**
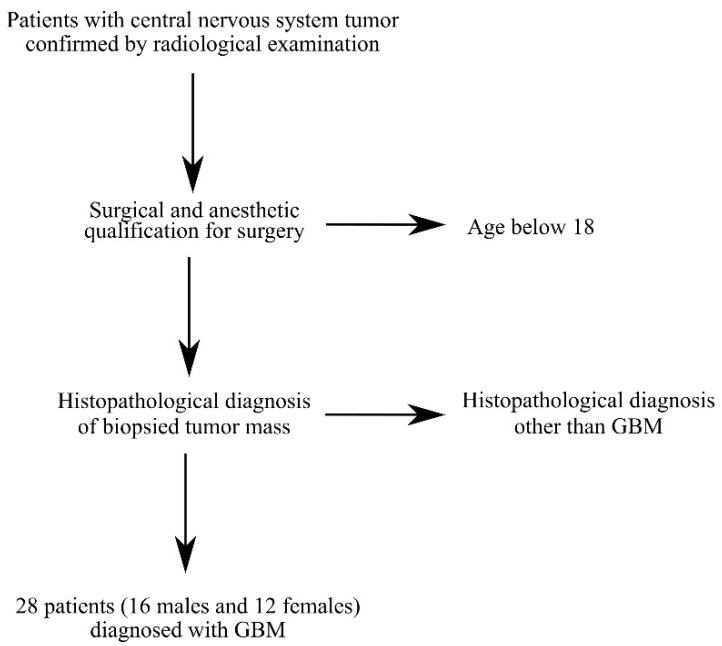
Criteria for selecting patients for the study. Patients were selected based on CNS tumor diagnosis, GBM diagnosis and age over 18 years.

**Figure 9 ijms-23-08536-f009:**
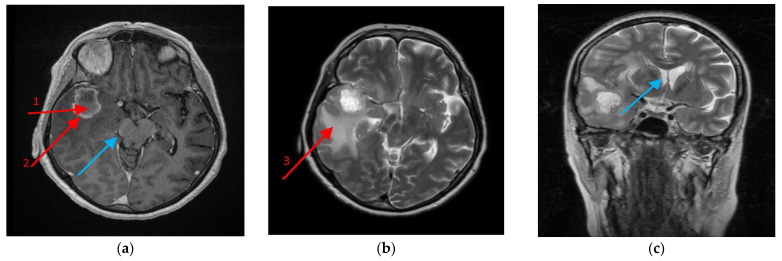
Magnetic resonance imaging (MRI) scan of the brain with the glioblastoma multiforme (GBM) tumor. A 64-year-old female with no antecedent medical history developed a headache, malaise and weight loss prior to hospital admission. On the brain MRI exam, a tumor was revealed in the right temporal lobe of 3–5 cm diameter with a **necrotic tumor core (1)** covered by a marginal **growing tumor area (2)** more dense in Clarisan, i.e., (**a**) On the T2 protocol, a tumor was seen as surrounded with an abundant oedematic area—**peritumoral area (3)**, (**b**) and caused a right brain peduncle compression (**a**) (blue arrow), as well as a lateral ventricle shift, (**c**) (blue arrow). The biopsy revealed GBM, which was then followed by craniotomy and the tumor was removed in gross totally by neuronavigation assistance.

**Table 1 ijms-23-08536-t001:** Statistical characteristics of the study group.

	N	Mean	Standard Deviation	Median	Minimum	Maximum	First Quartile	Third Quartile	InterQuartile Range
Age at surgery	24	60.7	12.5	64	36	81	54	68.5	14.5
Weight	24	84	19	89	55	130	67.5	95	27.5
Height	23	172	12	172	147	196	163	182	19
BMI	23	28.7	4.8	27.9	21.5	38.9	24.7	31.9	7.2
Physical activity	21	3.05	1.16	3	1	4	3	4	1
Limitation of physical activity caused by the tumor	21	2.10	0.83	2	1	3	1	3	2
Limitation of cognitive abilities	21	2.19	0.87	2	1	3	1	3	2

N, number of patients included in the analysis. Physical activity, limitation of physical activity and limitation of cognitive abilities were calculated based on the levels indicated in the questionnaires. Physical activity: everyday, 4; a few times a week, 3; rarely, 2; almost never, 1. Limitation of physical activity: none, 1; partial, 2; considerable, 3. Limitation of cognitive abilities: none, 1; partial, 2; considerable, 3.

**Table 2 ijms-23-08536-t002:** Statistical characteristics of the study group.

	N	Mean	Standard Deviation	Median	Minimum	Maximum	First Quartile	Third Quartile	InterQuartile Range
	**Men**								
Age at surgery	14	60.6	11.9	62	41	81	57.5	67	9.5
Weight	14	93.8	15.4	93	73	130	82.5	97.6	15.1
Height	14	178	8.6	178	163	196	172	184	12
BMI	14	29.5	4.2	28.0	24.7	38.9	27.0	31.7	4.6
Physical activity	12	3.17	1.19	4	1	4	2.75	4	1.25
Limitation of physical activity caused by the tumor	12	2.33	0.78	2.5	1	3	2	3	1
Limitation of cognitive abilities	12	2.5	0.67	3	1	3	2	3	1
	**Women**								
Age at surgery	12	60.8	14.1	66	36	79	53.5	70.2	16.8
Weight	12	70.3 **	15	63.5	55	95	59	83.5	24.5
Height	11	162 **	8	160	147	173	158	168	10
BMI	11	27.4 *	5.7	25.4	21.5	36.2	22.2	31.9	9.7
Physical activity	11	2.89 **	1.17	3	1	4	3	4	1
Limitation of physical activity caused by the tumor	11	1.78 **	0.83	2	1	3	1	2	1
Limitation of cognitive abilities	11	1.78 **	0.97	1	1	3	1	3	2

N, number of patients included in the analysis. Physical activity, limitation of physical activity and limitation of cognitive abilities were calculated based on the levels indicated in the questionnaires. Physical activity: everyday, 4; a few times a week, 3; rarely, 2; almost never, 1. Limitation of physical activity: none, 1; partial, 2; considerable, 3. Limitation of cognitive abilities: none, 1; partial, 2; considerable, 3. The difference between men and women statistically was significant at ** *p* < 0.001; * *p* < 0.05.

## Data Availability

The data presented in this study are available on request from the corresponding author.

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
