# Peer review of "CCL18 Expression Is Higher in a Glioblastoma Multiforme Tumor than in the Peritumoral Area and Causes the Migration of Tumor Cells Sensitized by Hypoxia"

_ijms, 2022, doi:10.3390/ijms23158536_

Round 1
Reviewer 1 Report
Dear authors:
I really liked your draft. However, it deserves to be improved. Please find below a few suggestions:
- A “limitations of the study” section would greatly benefit the quality of the discussion and would help the reader understanding the potential of the results presented.
- Abstract, line 10. “Obserwed”, needs respelling.
- Though a limited number of patients is recruited I recommend analyzing and stating the statistical significance of the demographic and basic characteristic of the study group regarding sex (Table 2).
- Reference bar in microphotographs should be added
Author Response
Reviewer 1
Comments and Suggestions for Authors
Dear authors:
I really liked your draft. However, it deserves to be improved. Please find below a few suggestions:
- A “limitations of the study” section would greatly benefit the quality of the discussion and would help the reader understanding the potential of the results presented.
We greatly appreciate your kind and constructive review. Thank you very much for this remark, we have added a “limitation of the study “ section.
- Abstract, line 10. “Obserwed”, needs respelling.
Corrected
- Though a limited number of patients is recruited I recommend analyzing and stating the statistical significance of the demographic and basic characteristic of the study group regarding sex (Table 2).
Thank you very much for this remark, we have added a statistical analysis for demographic and basic characteristic of the study group
- Reference bar in microphotographs should be added
Corrected according to Reviewer remark
Reviewer 2 Report
The authors investigated the expression of CCL18 under hypoxic conditions in GBM. Authors supported their hypothesis using patient samples and showed increased CCL18 expression in the core of tumors suggesting a possible link between hypoxia and CCL18 expression. As CCL18 is a critical mediator for the homeostasis of the tumor microenvironment this study presents an important perspective. My comments for the authors are below.
Major Comments;
Gene expression analysis result in figure 3 shows very minor difference between the groups but it is still statistically significant. Can authors comment if the experiment is derived from biological replicates? If so how many.
In order to show if the effect of hypoxia on cell increased cell migration is CCL18 dependent, authors should do the CoCl2 treatments with and without siRNA or shRNA targeting PITPNM3 and CCR8 receptors .
Minor comments;
In Figures 6 and 7, scale bars are missing.
Some paragraphs can be linked each other better.
Author Response
Reviewer 2
The authors investigated the expression of CCL18 under hypoxic conditions in GBM. Authors supported their hypothesis using patient samples and showed increased CCL18 expression in the core of tumors suggesting a possible link between hypoxia and CCL18 expression. As CCL18 is a critical mediator for the homeostasis of the tumor microenvironment this study presents an important perspective. My comments for the authors are below.
Major Comments;
Gene expression analysis result in figure 3 shows very minor difference between the groups but it is still statistically significant. Can authors comment if the experiment is derived from biological replicates? If so how many.
We greatly appreciate your kind and constructive review.
Cell culture and brain tumor analysis were performed in 6 replications for each study group.
In order to show if the effect of hypoxia on cell increased cell migration is CCL18 dependent, authors should do the CoCl2 treatments with and without siRNA or shRNA targeting PITPNM3 and CCR8 receptors .
Thank you very much for this comment of the reviewer. We agree that such research, i.e. CoCl2 treatments with and without siRNA or shRNA targeting PITPNM3 and CCR8 receptors, will be very useful. Unfortunately, we do not currently have such a research technique. However, we will use this suggestion in our further research if we obtain further research funding. We've added this explanation in the limitation of the study.
Minor comments;
In Figures 6 and 7, scale bars are missing.
According to the reviewer's comment, the figures have been corrected. The scale has been added.
Some paragraphs can be linked each other better.
Thank you for this remark, we've tried to improve the entire manuscript in this regard.
Round 2
Reviewer 2 Report
Thank you for addressing my points.
Author Response
Thank you very much for this remark. We are sending a revised "Limitation of the study section". We hope that we fulfils requirements of Reviewier.
Limitation of the study
This study showed that macrophages secrete CCL18, which increases the migration of GBM cells sensitized by hypoxia. It also demonstrated gender differences in the expression of receptors for CCL18. Nevertheless, it was associated with certain limitations. First, this study considered patients from a population from a relatively small region (Poland) that is relatively genetically homogeneous so the obtained results may differ from a similar study conducted elsewhere. For this reason, the experiment should be replicated in other countries to see whether its results were specifically related to the studied population or GBM in general. Second, the mechanism by which hypoxia sensitizes GBM cells to CCL18 should be investigated in greater detail. Our results indicate that hypoxia increases the expression of PITPNM3, the receptor for CCL18, which points to a potential mechanism in which hypoxia sensitizes GBM cells to CCL18 by increasing PITPNM3 expression. However, we did not focus on the importance of the PITPNM3 receptor in this process, but only on the potential role of CCL18 in GBM tumorigenesis. For this reason, future research should investigate the importance of PITPNM3 in GBM cell migration under hypoxia. Particularly useful could be experiments using siRNA or shRNA targeting PITPNM3 and CCR8 receptors.
